# A Rare Tetrad of Sickle Cell Disease, Vascular Ehlers–Danlos Syndrome, Primary Ciliary Dyskinesia, and Phelan–McDermid Syndrome in a Saudi Child: A Complex Multisystem Pediatric Case Report

**DOI:** 10.3390/pediatric17050089

**Published:** 2025-09-04

**Authors:** Gassem Gohal

**Affiliations:** Department of Pediatrics, Faculty of Medicine, Jazan University, P.O. Box 2097, Jazan 45142, Saudi Arabia; dr.gassem@gmail.com

**Keywords:** sickle cell disease, vascular Ehlers-Danlos syndrome, primary ciliary dyskinesia, Phelan-McDermid syndrome, coexistence of genetic disorders, consanguinity

## Abstract

Background: The coexistence of sickle cell disease (SCD), vascular Ehlers–Danlos syndrome (vEDS), primary ciliary dyskinesia (PCD), and Phelan–McDermid syndrome (PMS) in a single pediatric patient is extremely rare and poses substantial diagnostic and management challenges. Case presentation: We report an 8-year-old male from Jazan, Saudi Arabia, born to consanguineous parents, with early-onset SCD, followed by the identification of vEDS, PCD, and PMS through clinical presentation and whole exome sequencing. His disease course has been exceptionally severe, marked by monthly hospitalizations, multiple PICU admissions, and a wide spectrum of systemic complications. Conclusions: The coexistence of SCD, vEDS, PCD, and PMS may lead to synergistic vascular, pulmonary, and neurodevelopmental compromise, demanding multidisciplinary long-term management. This case underscores the need for a comprehensive targeted genetic assessment in patients with unusually aggressive or syndromic SCD phenotypes, particularly in regions with high levels of consanguineous marriages.

## 1. Introduction

Sickle cell disease (SCD) is one of the most prevalent and debilitating autosomal recessive hemoglobinopathies, particularly in the pediatric population of the Jazan region, Saudi Arabia [1].

Globally, SCD affects more than 20 million individuals, with sub-Saharan Africa and the Middle East representing major endemic areas. In Saudi Arabia, particularly in the Eastern and Southwestern regions, prevalence reaches up to 145 cases per 10,000 births, highlighting its major public health impact. This underscores the significance of early diagnosis and multidisciplinary care in these populations [2,3], where the disease burden is notably high due to consanguinity and genetic clustering. Clinically, SCD is characterized by chronic hemolytic anemia, intermittent vaso-occlusive crises, stroke, acute chest syndrome, and progressive vascular damage leading to multiorgan dysfunction over time [4]. SCD is caused by a single point mutation in the β-globin gene (HBB) located on chromosome 11p15.5, where adenine (A) is replaced by thymine (T) in codon 6 (GAG → GTG), resulting in the substitution of glutamic acid with valine at position six of the β-globin chain (p.Glu6Val). This structural change leads to hemoglobin S (HbS) polymerization under hypoxic conditions and the characteristic sickling of red blood cells [5].

Current therapeutic strategies include hydroxyurea, which reduces the frequency of crises and transfusion requirements by increasing fetal hemoglobin levels [6]. Regular blood transfusions are indicated in select patients to prevent stroke and manage severe anemia. Recently, bone marrow transplantation (BMT) has emerged as the only established curative treatment, with long-term survival rates exceeding 90% in matched sibling donor settings [7]. However, availability, donor compatibility, and transplant-related risks limit its use to a minority of eligible patients. Recently, gene therapy has emerged as a transformative treatment for sickle cell disease. In December 2023, the U.S. FDA approved two autologous stem cell-based therapies: exagamglogene autotemcel (Casgevy), a CRISPR/Cas9 genome-editing approach that reactivates fetal hemoglobin, and lovotibeglogene autotemcel (Lyfgenia), a lentiviral vector-mediated therapy introducing a modified β-globin gene. Clinical trials have shown that the majority of treated patients remained free of vaso-occlusive crises, marking a major advance in curative options for SCD [8,9]. The recent introduction of national newborn screening for sickle cell disease, combined with the longstanding premarital screening and genetic counseling programs, offers a promising framework for early diagnosis, interventions, and family planning—especially in high-prevalence areas like Jazan

Ehlers–Danlos syndromes (EDSs) represent a diverse group of inherited connective tissue disorders, mostly inherited in an autosomal dominant manner, while rare subtypes follow an autosomal recessive pattern. The overall prevalence of EDS is estimated at approximately 1 in 5000 individuals worldwide, although the vascular subtype (vEDS) is much rarer, with an estimated prevalence of 1 in 100,000 to 200,000. EDSs primarily affect the skin, joints, and blood vessels. Clinically, EDS is characterized by features such as joint hypermobility, skin hyperextensibility, tissue fragility, and a tendency toward easy bruising [10]. To date, 13 distinct EDS subtypes have been identified, reflecting the growing recognition of its clinical and genetic heterogeneity [11]. Among the 13 currently recognized subtypes, vascular EDS (vEDS) is one of the most severe and is caused by heterozygous pathogenic variants in the COL3A1 gene, located on chromosome 2q32.2. This gene encodes type III procollagen, essential for the integrity of vascular walls. Mutations result in defective or deficient collagen, leading to fragile connective tissue and a high risk of spontaneous arterial or organ rupture [12]. The clinical presentation varies significantly, ranging from mild joint laxity and skin findings to disabling musculoskeletal symptoms and potentially life-threatening vascular events (arterial dissection, Aneurism, or organ rupture) [13]. In addition to physical symptoms, neurological and psychiatric features have been increasingly reported. These include spinal instability, concentration difficulties, and neurocognitive deficits affecting attention, memory, and visuospatial functioning [14,15].

Primary ciliary dyskinesia (PCD) is a rare autosomal recessive disorder with an estimated global prevalence ranging from 1 in 10,000 to 1 in 30,000 individuals. In populations with high rates of consanguinity, such as in the Middle East, the prevalence is likely higher. In Saudi Arabia, only a limited number of cases have been reported, reflecting both the rarity of the condition and challenges in diagnosis [16]. Resulting from mutations in more than 50 genes involved in the structure and function of motile cilia, including DNAI1, DNAH5, RSPH9, and CCDC39 [17,18], these mutations disrupt effective ciliary movement, leading to impaired mucociliary clearance.

Clinically, PCD often manifests in the neonatal period with unexplained respiratory distress in full-term infants, followed by a persistent, chronic wet cough, recurrent otitis media, and nasal congestion in early childhood. Approximately 50% of patients exhibit situs inversus, and this triad (chronic sinusitis, bronchiectasis, and situs inversus) defines Kartagener syndrome [19]. The disease course is typically chronic and progressive, with recurrent lower respiratory tract infections that can lead to bronchiectasis, declining lung function, and male infertility due to immotile sperm [20]. Diagnosis involves nasal nitric oxide measurement, high-speed video microscopy, electron microscopy, and molecular genetic testing [21]. While there is currently no definitive cure, comprehensive management—including airway clearance therapy, prophylactic antibiotics, treatment of exacerbations, and multidisciplinary pulmonary care—has been shown to slow progression and improve quality of life [22].

Phelan–McDermid syndrome (PMS) is an autosomal dominant neurodevelopmental disorder; its prevalence is estimated at approximately 1 in 8000 to 1 in 15,000 individuals, although the true frequency may be higher due to underdiagnosis and limited access to genetic testing. Primarily caused by the terminal deletion of chromosome 22q13.3 or pathogenic variants in the SHANK3 gene, which encodes a postsynaptic density protein critical for synaptic development and function, the estimated incidence ranges from 1 in 8000 to 15,000 live births. Clinically, PMS presents with neonatal hypotonia, global developmental delay, absent or severely delayed speech, intellectual disability, and autism spectrum disorder features. Other manifestations may include dysmorphic facial features, seizures, gastrointestinal disturbances, and renal anomalies. Diagnosis is confirmed through chromosomal microarray or SHANK3 sequencing. Currently, there is no curative therapy; management is supportive and includes early intervention services such as physical, speech, behavioral, and occupational therapies. The prognosis varies based on deletion size and comorbidities, though most affected individuals require long-term multidisciplinary care and educational support [23].

While each of these disorders has been reported individually or occasionally in dual combinations, to our knowledge, the coexistence of four genetically distinct syndromes—SCD, vEDS, PCD, and PMS—in a single patient has not previously been documented. This rare concurrence creates a compounded clinical phenotype that is more severe than any of the conditions alone.

## 2. Case Presentation

A male child, born in June 2016 to consanguineous Saudi parents from the Jazan region, was diagnosed with sickle cell anemia (SCD) at 3 months of age following episodes of dactylitis and severe anemia. Pregnancy and perinatal history were unremarkable, with normal antenatal ultrasounds and no perinatal asphyxia. Apgar scores were within the normal range, and birth weight was at the 25th percentile. Family pedigree analysis showed that both parents were heterozygous for the S allele and clinically asymptomatic for vEDS, PCD, or PMS. Two siblings were affected with SCD, but none of them demonstrated a similar complex presentation as observed in our patient. The recurrence risk of additional affected children was discussed with the parents, and formal genetic counseling was provided. Notably, this patient uniquely presented with the co-occurrence of four distinct genetic disorders.

At 8 months of age, he began exhibiting clinical features suggestive of a connective tissue disorder, including soft, hyperextensible skin and increased joint laxity. Additionally, he displayed global developmental delay, subtle dysmorphic features, and bilateral undescended testes. Notably, he was born with a right knee dislocation, which was successfully reduced during the neonatal period and was retrospectively considered a musculoskeletal manifestation of Ehlers–Danlos syndrome (EDS). On clinical examination, the child exhibited generalized hypotonia, marked ligamentous laxity, and craniofacial dysmorphic features, including hypertelorism and downslanting palpebral fissures. His growth parameters were consistently below the 25th percentile for age. A cardiovascular assessment revealed mild mitral regurgitation, and an abdominal ultrasound demonstrated splenomegaly prior to his splenectomy in 2022. Pulmonary CT imaging showed early bronchiectatic changes compatible with primary ciliary dyskinesia (PCD). Neuroimaging with a brain MRI excluded overt cerebral infarction but revealed nonspecific white matter changes, while magnetic resonance angiography (MRA) demonstrated no evidence of cerebrovascular stenosis or moyamoya-like vasculopathy at the time of evaluation.

Whole exome sequencing identified four significant pathogenic variants: a homozygous HBB mutation (c.20A > T) consistent with SCD; a heterozygous COL3A1 mutation diagnostic of vascular EDS (type IV); a homozygous RSPH9 mutation confirming primary ciliary dyskinesia (PCD); and a heterozygous pathogenic SHANK3 variant, establishing the diagnosis of Phelan–McDermid syndrome (PMS).

The patient has experienced a markedly severe clinical course. He required near-monthly hospitalizations for blood transfusions during early childhood, with some clinical stabilization noted only after undergoing splenectomy in June 2022. He has been admitted to the pediatric intensive care unit (PICU) multiple times due to severe anemia, hemolytic crises, and acute chest syndrome, with hemoglobin levels documented as low as 2 g/dL. He was also diagnosed with and treated for osteomyelitis on three separate occasions and underwent multiple exchange transfusions in the context of recurrent acute chest syndrome (Figure 1).

Recurrent episodes of shortness of breath necessitated inhaled corticosteroid therapy; however, treatment efficacy has been compromised by generalized hypotonia and ligamentous laxity, likely attributable to the combined impact of EDS and PMS, complicating proper inhaler technique. Neurodevelopmentally, the patient continues to exhibit global developmental delays, significant speech and learning difficulties, and visual impairment requiring corrective lenses. These findings are in keeping with the clinical phenotype of Phelan–McDermid syndrome, which is known to involve intellectual disability, hypotonia, and features of autism spectrum disorder.

Current treatment includes hydroxyurea, folic acid supplementation, inhaled short-acting beta-agonists, and corticosteroids for respiratory symptoms, as well as regular physiotherapy to address hypotonia and joint instability. He receives academic and speech-language support, and care is coordinated through a multidisciplinary team comprising specialists in genetics, hematology, pulmonology, urology, neurology, social work, and rehabilitation.

At present, the child remains medically complex and hospitalized due to ongoing joint pain, recurrent respiratory complications, and developmental challenges reflective of the compounded burden of SCD, vEDS, PCD, and PMS. Furthermore, the duration and severity of illness and the time of recovery from each crisis takes longer periods in comparison to a patient affected with SCD without similar comorbidities.

## 3. Discussion

To our knowledge, this is the first reported pediatric case involving the coexistence of four genetically distinct disorders: sickle cell disease (SCD), vascular Ehlers–Danlos syndrome (vEDS), primary ciliary dyskinesia (PCD), and Phelan–McDermid syndrome (PMS). Each of these conditions is independently rare and arises from mutations in separate, non-overlapping genes—HBB on chromosome 11p15.5 (SCD), COL3A1 on 2q32.2 (vEDS), RSPH9 on 6p21 (PCD), and SHANK3 on 22q13.3 (PMS). The absence of known molecular or regulatory interactions among these genes suggests that this combination likely results from chance co-inheritance in the context of consanguinity, which is a known risk factor in certain populations, including the Jazan region of Saudi Arabia.

For the two dominant conditions (vEDS and PMS), both parents were clinically asymptomatic and were not tested, as confirmatory genetic testing is not routinely available in our setting. This raises the possibility of de novo mutations. As approximately half of vascular EDS cases and the majority of SHANK3-related PMS cases arise from de novo variants [12,22], it is plausible that the dominant mutations identified in our patient occurred as de novo events, although this cannot be confirmed in the absence of parental testing. While each of these genetic disorders presents considerable clinical challenges on their own, their co-occurrence in this patient has resulted in a synergistic and compounded disease burden. The clinical phenotype observed in this child significantly surpasses the severity typically seen in isolated sickle cell disease and cannot be fully explained by SCD alone.

SCD is characterized by recurrent vaso-occlusive crises (VOCs), hemolytic anemia, and progressive end-organ damage driven by hemoglobin S polymerization, erythrocyte sickling, and chronic endothelial dysfunction. These processes lead to ischemic injury in multiple organ systems, particularly the lungs, spleen, and brain [24].

Superimposed on this hematologic burden is vascular Ehlers–Danlos syndrome, resulting from mutations in COL3A1, which impair the synthesis of type III collagen—an essential component of vascular and organ wall integrity. This structural deficiency leads to arterial fragility, the increased risk of spontaneous rupture, and poor wound healing [11]. The combination of SCD-induced ischemia and vEDS-associated tissue fragility creates a particularly dangerous vascular phenotype, where even minor vascular insults can precipitate life-threatening complications [25].

Primary ciliary dyskinesia further compounds the patient’s condition by contributing chronic respiratory dysfunction due to defective mucociliary clearance. The homozygous RSPH9 mutation identified in this patient impairs effective ciliary movement, leading to recurrent respiratory infections, persistent airway inflammation, and bronchiectasis [26]. In children with SCD, pulmonary infections are a well-established trigger for acute chest syndrome (ACS). The co-occurrence of PCD in this case may serve as a continuous pro-inflammatory stimulus, heightening the frequency and severity of ACS episodes and necessitating frequent hospitalizations and exchange transfusions.

Adding to this already complex triad is Phelan–McDermid syndrome (PMS), caused by a pathogenic variant in the SHANK3 gene. SHANK3 encodes a synaptic scaffolding protein vital for neuronal development and plasticity. Mutations or deletions in this gene are associated with global developmental delay, hypotonia, intellectual disability, severely delayed or absent speech, and features of autism spectrum disorder [23]. In this patient, PMS contributes significant neurodevelopmental morbidity, including cognitive delay, hypotonia, impaired communication, and poor compliance with medical interventions. Importantly, the hypotonia and ligamentous laxity associated with PMS and vEDS negatively affect respiratory mechanics and compromise inhaler use, further exacerbating the pulmonary complications from PCD and SCD.

The combined presence of SCD, vEDS, PCD, and PMS creates a uniquely complex, multisystem phenotype in which hematologic, vascular, respiratory, and neurologic systems are simultaneously and progressively impaired. This “quadruple-pathology burden” transforms what might otherwise be a manageable monogenic disorder into a medically fragile syndrome that requires individualized, multidisciplinary care. The patient’s early-onset complications, high transfusion needs, frequent PICU admissions, developmental delays, and respiratory decompensations all reflect the synergistic interplay among these disorders.

From a pathophysiological perspective, this case highlights how overlapping genetic defects can act synergistically to accelerate organ dysfunction. For instance, recurrent hypoxia and hemolysis in SCD increase oxidative stress, which may further destabilize fragile vascular structures in vEDS. Similarly, the chronic inflammatory state induced by PCD not only predisposes one to acute chest syndrome but may also worsen endothelial dysfunction, amplifying the risks already present in SCD. Neurodevelopmental limitations from PMS further reduce adherence to medical interventions, creating a vicious cycle of poor disease control. Comparable reports in the literature usually describe the coexistence of two rare diseases, such as SCD with connective tissue disorders or with neurodevelopmental conditions. However, the presence of four distinct genetic disorders in a single patient has not been described before. This suggests that in populations with high rates of consanguinity, the probability of inheriting multiple pathogenic variants is significantly increased, underscoring the urgent need for strengthened genetic counseling and premarital screening programs. The management of such a patient is highly challenging. Standard therapies for SCD, such as hydroxyurea, transfusions, and bone marrow transplantation, must be considered in light of increased surgical and vascular risks from vEDS. Similarly, physiotherapy and inhaled therapies for PCD are hindered by hypotonia and developmental delays in PMS. Thus, optimal care requires not only multidisciplinary coordination but also the individualized adaptation of treatment protocols to address overlapping limitations. From a public health standpoint, this case has important implications. It reinforces the need for comprehensive newborn screening programs that extend beyond SCD to include broader genetic panels, particularly in high-risk regions. Moreover, it emphasizes the importance of establishing rare disease registries and collaborative research networks to capture such unusual phenotypes, improve knowledge sharing, and guide the development of targeted therapeutic approaches.

This case underscores the critical importance of comprehensive genetic evaluations in children who present with atypical or disproportionately severe manifestations of a known genetic condition such as SCD. The early identification of coexisting genetic disorders can profoundly impact prognosis, guide therapeutic interventions, and support tailored family counseling—particularly in regions with high rates of consanguinity.

## 4. Conclusions

This case highlights the unprecedented co-occurrence of four genetically distinct disorders—sickle cell disease (SCD), vascular Ehlers–Danlos syndrome (vEDS), primary ciliary dyskinesia (PCD), and Phelan–McDermid syndrome (PMS)—in a single pediatric patient. Each condition independently contributes to significant morbidity, yet their simultaneous presence creates a compounded multisystem pathology involving the hematologic, vascular, pulmonary, and neurodevelopmental domains. The resulting phenotype is not merely additive but synergistically severe, leading to frequent hospitalizations, intensive care admissions, and a rapid progression of clinical complications.

This case underscores the importance of comprehensive genetic testing in children presenting with atypical or disproportionately severe features of a known genetic disorder such as SCD, particularly in populations with high rates of consanguinity. Early diagnosis and multidisciplinary management are essential to optimizing outcomes and providing accurate genetic counseling to affected families. To our knowledge, this is the first reported pediatric case combining SCD, vEDS, PCD, and PMS. It highlights the need for comprehensive genetic evaluations in children with unusually severe presentations. In regions with high consanguinity and genetic disease prevalence, early diagnosis is vital. Identifying complex syndromes early may enable proactive care and improve long-term outcomes in high-risk patients.

## Figures and Tables

**Figure 1 pediatrrep-17-00089-f001:**
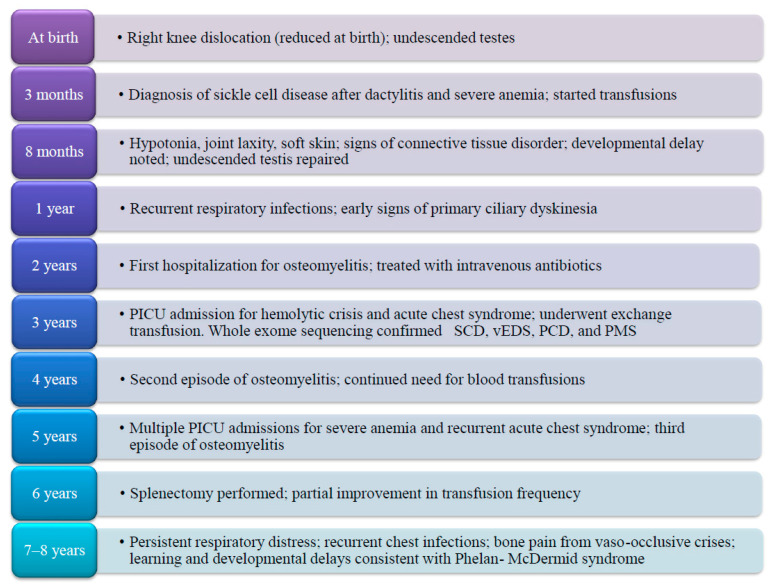
Timeline and summary of case presentation, therapeutic interventions, and follow-up of a pediatric case affected with a rare tetrad of sickle cell disease, vascular Ehlers–Danlos syndrome, primary ciliary dyskinesia, and Phelan–McDermid syndrome. PICU: pediatric intensive care unit.

## Data Availability

The data presented in this study are available on request from the corresponding author due to patient privacy concerns.

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
