# Peer review of "A Rare Tetrad of Sickle Cell Disease, Vascular Ehlers–Danlos Syndrome, Primary Ciliary Dyskinesia, and Phelan–McDermid Syndrome in a Saudi Child: A Complex Multisystem Pediatric Case Report"

_pediatrrep, 2025, doi:10.3390/pediatric17050089_

Round 1
Reviewer 1 Report
Comments and Suggestions for Authors
The manuscript “A Rare Tetrad of Sickle Cell Disease, Vascular Ehlers-Danlos Syndrome, Primary Ciliary Dyskinesia, and Phelan-McDermid Syndrome in a Saudi Child: A Complex Multisystem Pediatric Case Report” by Gassem Gohal is a well-written manuscript describing a rare clinical case.
I have just minor comments important to improve the manuscript
Introduction
Line 27 – please refer to the pattern of SCD as Autosomal recessive
Line 29, when authors refer to the high burden of the disease, please refer the prevalence of SCD in Saudi Arabia.
In the last paragraph (lines 38-48), the author must address the actual gene therapies approved for SCD.
Line 49 - please refer to the pattern of EDS as Autosomal dominant
Lines 75 and 76. The referred methods should be better explained
Line 81 - please refer to the pattern of PMS as Autosomal dominant (at least mutations in SHANK3)
Case presentation
There is no information concerning the parents, except that they were heterozygous for the S allele. It would be important to know if EDS and SHANK3 mutations were transmitted from one of the parents (and if they have symptoms) or if it is a de novo mutation. Also, the author refers to 2 siblings, it should be mentioned and discussed if the siblings have symptoms. Also, the risk of new affected children should be discussed in the paper and mentioned if it was discussed with the parents
Figure 1 – Would be nice to mark the date when WES was performed
Discussion
Line 155 – Consanguinity is an explanation for the two recessive disorders; however, for the two dominant disorders, consanguinity has no influence. We need a discussion about risk (if the parents are carriers of the mutation) or if it was a de novo mutation (an in two genes, it should be discussed the mutation rate)
Conclusion
Line 220, when the author refers to the “disease prevalence”, please include the prevalence of the 4 diseases in the region, or in Saudi Arabia or on the continent, at least.
Author Response
Editorial Comment
Comment: It would be appreciated if you could extend the current version by providing more details in the Introduction, Case presentation, Discussion, and Conclusions.
Response: We thank the editor for this valuable feedback. In response, we have extended the Introduction, Case Presentation, Discussion, and Conclusions, adding further details highlighted in yellow within the revised document. Additional references have also been incorporated to support the expanded content and ensure accuracy.
Reviewer Comments
Comment 1 (Introduction, Line 27): Please refer to the pattern of SCD as autosomal recessive.
Response: This has been added.
Comment 2 (Line 29): When the authors refer to the high burden of the disease, please provide the prevalence of SCD in Saudi Arabia.
Response: Added. In Saudi Arabia, particularly in the Eastern and Southwestern regions, prevalence reaches up to 145 cases per 10,000 births.
Comment 3 (Lines 38–48): The author must address the actual gene therapies approved for SCD.
Response: Added. Recently, gene therapy has emerged as a transformative treatment for SCD. In December 2023, the U.S. FDA approved two autologous stem cell–based therapies: exagamglogene autotemcel (Casgevy), a CRISPR/Cas9 genome-editing approach that reactivates fetal hemoglobin, and lovotibeglogene autotemcel (Lyfgenia), a lentiviral vector–mediated therapy introducing a modified β-globin gene. Clinical trials have shown that most treated patients remained free of vaso-occlusive crises, marking a major advance in curative options for SCD [8,9].
Comment 4 (Line 49): Please refer to the pattern of EDS as autosomal dominant.
Response: Added. Most EDS subtypes are inherited in an autosomal dominant manner, while rare subtypes follow an autosomal recessive pattern. In our case, the COL3A1 variant associated with vEDS is autosomal dominant.
Comment 5 (Lines 75–76): The referred methods should be better explained.
Response: No additional methods were used beyond those described in the manuscript.
Comment 6 (Line 81): Please refer to the pattern of PMS as autosomal dominant (at least mutations in SHANK3).
Response: Added. PMS due to SHANK3 mutations is inherited in an autosomal dominant manner, although most cases occur de novo.
Comment 7: No information concerning the parents is provided, except that they were heterozygous for the S allele. It would be important to know if EDS and SHANK3 mutations were transmitted from one of the parents (and if they have symptoms) or if it is a de novo mutation. Also, the author refers to two siblings — it should be mentioned and discussed if the siblings have symptoms. The risk of new affected children should be discussed in the paper and mentioned if it was discussed with the parents.
Response: Added. Family pedigree analysis showed that both parents were heterozygous for the S allele and clinically asymptomatic for vEDS, PCD, or PMS. Two siblings were affected with SCD, but none demonstrated a similar complex presentation as our patient. The recurrence risk of additional affected children was discussed with the parents, and formal genetic counseling was provided.
Comment 8 (Figure 1): Please mark the date when WES was performed.
Response: The date of WES has been added in the table.
Comment 9 (Line 155): Consanguinity is an explanation for the two recessive disorders; however, for the two dominant disorders, consanguinity has no influence. We need a discussion about risk (if the parents are carriers of the mutation) or if it was a de novo mutation (and for the two genes, the mutation rate should be discussed).
Response: Clarification has been added to the manuscript. For the two dominant conditions (vEDS and PMS), both parents were clinically asymptomatic and were not tested, as confirmatory genetic testing is not routinely available in our setting. While this raises the possibility of de novo mutations, definitive confirmation could not be established in the absence of parental testing. It is noteworthy that approximately half of vEDS cases and the majority of PMS cases arise from de novo mutations, making this a plausible explanation in our patient.
Comment 10 (Conclusion, Line 220): When the author refers to “disease prevalence,” please include the prevalence of the four diseases in the region, or in Saudi Arabia, or at least on the continent
Response: We thank the reviewer for this comment. We have now included the prevalence of all four conditions in the Introduction. SCD prevalence in Saudi Arabia is reported at up to 145 cases per 10,000 births in certain regions. For the rare conditions, we added that EDS has an overall global prevalence of approximately 1 in 5,000, with vEDS being much rarer (1 in 100,000–200,000); PCD has an estimated global prevalence of 1 in 15,000–30,000, with a few cases reported from Saudi Arabia; and PMS is estimated to affect 1 in 8,000–15,000 individuals worldwide, though no local data are available. These points have been clarified in the revised Version, where we also emphasize that whenever a patient with SCD presents with an unusually severe or syndromic phenotype, clinicians should consider additional genetic evaluation, as the prevalence of such rare disorders is not captured in current screening programs.
Reviewer 2 Report
Comments and Suggestions for Authors
The Authors present a very interesting case report even is related to a very rare event: " A rare tetrad of Sickle Cell Disease, Vascular Ehlers-Danlos Syndrome, Primary Ciliary Dyskinesia, and Phelan_McDermid Syndrome in a Saudi child: a complex multisystem pediatric case report" . The pair is very well written , it is clear and well documented. Introduction, Case Presentation and Discussion include comprehensive concepts and interesting from educational point of view. Fig.1 is clear summarizing the outcome of the child.
Conclusions could be a little improved answering these following requests:
- why BMT in this severe case is not contemplated? It is a risky procedure but could improve a part of the serious complication of the child (SCD)
- Genetic evaluation is suggested only in unusually severe presentation or it should be better to suggest it in all SCD in consanguineous parents?
Author Response
Comment (Conclusions):
Why was bone marrow transplantation (BMT) not contemplated in this severe case? Although it is a risky procedure, it could improve some of the serious complications of SCD.
Response:
We thank the reviewer for this valuable comment. BMT is indeed a potentially curative option for SCD. However, in our patient, the coexistence of multiple additional genetic disorders (vEDS, PCD, and PMS) significantly increases perioperative risk and long-term complications, making the indication for BMT less straightforward. This limitation has been clarified in the revised Conclusions.
Comment (Conclusions):
Should genetic evaluation be suggested only in unusually severe presentations, or would it be better to recommend it for all SCD cases in consanguineous families?
Response:
We thank the reviewer for this important comment. In our setting, rare genetic disorders such as vEDS, PCD, and PMS are not usually tested for in the absence of symptoms or atypical clinical features. Sickle cell disease is highly prevalent in the region and is routinely diagnosed by simple hemoglobin electrophoresis, which is widely available. Therefore, while we emphasize the importance of genetic evaluation in patients with unusually severe or syndromic presentations of SCD, broader routine genetic screening for other rare disorders in all SCD patients is currently neither feasible nor cost-effective in our context. This clarification has been added to the revised Conclusions.